# Modulation of Heat-Shock Proteins Mediates Chicken Cell Survival against Thermal Stress

**DOI:** 10.3390/ani10122407

**Published:** 2020-12-16

**Authors:** Abdelrazeq M. Shehata, Islam M. Saadeldin, Hammed A. Tukur, Walid S. Habashy

**Affiliations:** 1Department of Animal Production, Faculty of Agriculture, Al-Azhar University, Cairo 11651, Egypt; abdelrazeq@azhar.edu.eg; 2Department of Dairy Science and Food Technology, Institute of Agricultural Sciences, Banaras Hindu University, Varanasi 221005, India; 3Department of Animal Production, College of Food and Agricultural Sciences, King Saud University, Riyadh 11451, Saudi Arabia; tukurhammeda@gmail.com; 4Department of Animal and Poultry Production, Damanhour University, Damanhour 22511, Egypt; walidh55@gmail.com

**Keywords:** heat stress, oxidative stress, HSPs, apoptosis, thermotolerance, cytoprotection

## Abstract

**Simple Summary:**

The impact of heat stress is a major challenge causing economical losses to the poultry industry. Heat stress exerts a damaging effect on physiological responses such as immunity, oxidative stress, and intestinal and muscular functions. In this review, we highlight the potential cytoprotective effects of heat-shock proteins to antagonize the adverse effects of heat shock, and how we can modulate their functions with pharmacological and nutritional interventions.

**Abstract:**

Heat stress is one of the most challenging environmental stresses affecting domestic animal production, particularly commercial poultry, subsequently causing severe yearly economic losses. Heat stress, a major source of oxidative stress, stimulates mitochondrial oxidative stress and cell dysfunction, leading to cell damage and apoptosis. Cell survival under stress conditions needs urgent response mechanisms and the consequent effective reinitiation of cell functions following stress mitigation. Exposure of cells to heat-stress conditions induces molecules that are ready for mediating cell death and survival signals, and for supporting the cell’s tolerance and/or recovery from damage. Heat-shock proteins (HSPs) confer cell protection against heat stress via different mechanisms, including developing thermotolerance, modulating apoptotic and antiapoptotic signaling pathways, and regulating cellular redox conditions. These functions mainly depend on the capacity of HSPs to work as molecular chaperones and to inhibit the aggregation of non-native and misfolded proteins. This review sheds light on the key factors in heat-shock responses for protection against cell damage induced by heat stress in chicken.

## 1. Introduction

The relentless stresses imposed by high ambient temperatures and the associated oxidative stress exacerbate several health problems in domestic animals, primarily commercial chicken. For example, heat stress contributes to the spread of diseases and reduces feed utilization due to disrupting intestinal mucosa integrity and barrier function. It also contributes to damaging to skeletal muscle cells, which have a major role in growth, and negatively affects the immune system, resulting in poor growth and great economic losses [1,2]. These processes increase the levels of heat-denatured and misfolded proteins, leading to cell injury and apoptosis. The cell has developed adaptive and effective mechanisms to counteract cellular insult induced by heat stress. Among these mechanisms, there is the enhancement of cell survival-related gene expression, and the early activation of some molecules that work as molecular chaperones and contribute to cell survival processes during heat stress, such as heat-shock factors (HSFs) and heat-shock proteins (HSPs) [3,4,5].

HSF1 and HSF3 are the two main members of the HSF family involved in the regulation of HSPs in chicken under heat-stress conditions. HSPs are classified according to their molecular weights. HSP27, HSP60, HSP70, and HSP90 are the most extensively researched among HSP families. HSP60 and HSP70 attach to recently synthesized polypeptides, preventing their aggregation and mediating their folding to the native state [3,4], whereas HSP90 is associated with proteins in older phases of development, and modifies their pattern [6]. HSP27 undergoes rapid phosphorylation during heat stress, which results in actin polymerization and stress-fiber formation [7]. Both HSP70 and HSP27 are essential in maintaining proteins’ solubility in specific compartments of the cell [8]. In endothelial populations, HSP70 and HSP27 were also implicated in the protection of cells against apoptosis [5].

HSPs include small HSPs (sHSPs) that are expressed in various tissue types of most organisms in response to various stressors, including heat, chemical, and physiological stress. The induction of HSPs is an essential endogenous protection mechanism. The upregulation of these molecules enhances cell stability and develops thermotolerance during heat-stress conditions. HSPs, well-conserved molecules of transcriptional regulators, are essential for maintaining cellular homeostasis via promoting cell survival and preventing apoptotic processes in various cell types [7]. Recent developments in the field of biomedical research have led to a renewed interest in HSPs due to their vital roles in thermal stress and cell survival during exposure to other stresses [9]. HSPs have both chaperone and protease functions, which protect cells from different stresses and play a key role in preventing misfolding, refolding damaged proteins, and targeting severely damaged proteins for subsequent refolding [10].

There is a growing trend of the therapeutic targeting of HSPs via nutritional and pharmacological agents towards alleviating heat stress in commercial chicken [11,12,13]. For example, Kamboh et al. reported that flavonoid supplementation (genistein and hesperidin) downregulated HSP70 mRNA in the skeletal muscle of heat-stressed broilers [14]. They suggested that this downregulation might be due to the structural and functional resemblance of genistein with estrogens. Naturalplant-derived compounds might modulate HSP expression by tuning their induction during stress conditions through their biological and pharmacological activities, including antioxidant, detoxification, and anti-inflammatory activities [15,16].

## 2. Heat Stress

Heat stress is one of the most challenging environmental stresses affecting domestic animal production, particularly commercial poultry, subsequently contributing to severe yearly economic losses (Figure 1) [17]. Heat stress adversely affects growth performance, inflammatory cytokines, antioxidant capacity, blood corticosterone concentration, respiratory rate, and malondialdehyde (MDA) level [18]. Chronic heat stress reduces body gain [3], carcass yields, and immune responses [19], and elevates the level of liver HSP70 mRNA expression [3] in broiler chickens. Moreover, heat stress can negatively impact cellular structure and function, leading to a failure of transcription processes, RNA processing, and cell membrane structure/function [20]. Due to the feather cover and lack of sweat glands in chicken, the combinations of high environmental temperature and humidity have an adverse effect on thermoregulatory processes and significantly impair heat dissipation [13,17]. Heat stress resulted in increased protein degradation [21], while other findings demonstrated that heat stress led to cell apoptosis [22,23].

Several researchers found an association between elevated temperature and oxidative damage in broiler chicken, which becomes increasingly severe in a temperature-dependent manner [20]. During the later stage of acute heat stress, the downregulation of avian uncoupling proteins increases oxidative stress, leading to mitochondrial dysfunction and tissue injury [24]. Mujahid et al. first reported that mitochondrial superoxide radical generation in the skeletal muscle of meat-type chicken increased in response to high ambient temperature [25]. Heat stress increases the production of mitochondrial H_2_O_2_ in the liver, levels of MDA in the liver and breast muscle, and concentration of lactic acid in breast muscles of chicken [26]. In the pectoralis major muscle of heat-stressed chicken, heat stress increases lipid peroxidation and protein oxidation and alters the activity of antioxidant enzymes [27]. Together, these studies indicated that breast muscle is more susceptible to oxidative stress under high ambient temperatures compared to other tissue types. Protein oxidation in the muscle could also alter the pH and subsequently impair meat quality. It is also likely that heat stress causes a rise in reactive oxygen species (ROS) production, which induces deregulated intracellular Ca^2+^ concentrations and leads to various biological disturbances that adversely affect the meat quality of broiler chickens [26]. Some studies also showed that exposure to thermal stress causes apoptosis via cytochrome c released from mitochondria with ROS-dependent mitochondrial p53 translocation, and increases intracellular Ca^2+^ overload and Bax mitochondrial translocation [28].

High temperature leads to alterations in intestinal permeability and local inflammation in poultry [2,29]. Heat stress was shown to increase serum concentrations of corticosterone and systemic inflammatory cytokines, TNF-α, and IL-2 [18]. In poultry, growing research has specifically demonstrated the negative effects of high ambient temperatures on the immune response. Heat stress also induces several immune defects in the broiler by affecting the maturation of T and B cells in lymphoid tissue [30].

Several studies showed an association between severe heat-stress-caused lesions and the expression of HSP70 in chicken [31,32]. The intestinal mucosa is very susceptible to heat-stress-induced injury. Research in this area showed that a variety of environmental stressors can enhance the expression of HSP70 [33,34,35,36]. In this respect, abnormal proteins, such as aggregated or denatured proteins, are probably the most common trigger for HSP expression in cells [37]. Damage to the intestinal mucosa leads to decreased feed efficiency, and suppressed nutrient transporters and immune responses, and subsequently facilitates luminal bacteria translocation and impairs growth performance [38]. Drastic consequences of heat stress were found in intestinal segments’ pathological injuries, mainly epithelium integrity and villus structure [39,40]. Exposure to high temperatures causes several defects in the intestinal epithelium, including damage to villus length and tip integrity, and increased cell shedding, which is thought to reduce epithelial barrier functions and increase vulnerability to bacterial translocation [39,41]. Heat stress induces apoptosis in intestinal epithelial cells by accumulated ROS and activates the lysosomal–mitochondrial apoptotic pathway which is involved in triggering cathepsin-B release [42]. Intestinal mucosa damage was also observed following increased lactic dehydrogenase (LDH) levels in heat-stressed chicken [43].

## 3. A brief History of HSPs

HSPs, induced by heat stress and several extrinsic environmental stressors, play a crucial role in cellular protection against deleterious effects [44]. HSPs act as molecular chaperones, i.e., they decrease the possibility that proteins interact with each other in an undesirable way via their involvement in protein folding, translocation, higher-order assembly, and degradation [44]. HSPs recognize and bind to non-native proteins, either due to protein functionality effects of stress or because protein peptides do not fully synthesize, fold, assemble, or localize to a proper cellular compartment [45].

The survival of a cell is intricately associated with its capability to maintain cellular components. Presumably, the essential functions of HSPs in coping with environmental stresses are irreversibly reducing the aggregation of non-native proteins, and then targeting these non-native or aggregated proteins for degradation and elimination from the cells [45]. HSPs can be categorized into several families dependent on their molecular mass, including HSP100, HSP90, HSP70, HSP60, HSP40, and sHSPs [46].

The transcriptional expression of genes encoding HSP families exhibits a massive increase in response to heat stress, indicating that these molecules present an arm of essential defense mechanisms against cellular injury [47]. The activation of various intracellular signaling pathways in response to hyperthermia results in HSP expression [45]. In vertebrates, the inducible expression of genes encoding HSPs is mediated by a family of heat-shock transcription factors (HSFs), among which HSF1 is required for inducing a heat-shock response (Figure 2). HSF acts as the molecular link between environmental stresses and HSP induction [48]. Therefore, the activation of HSFs leads to translocating HSPs into the nucleus, binding to heat-shock elements (HSEs), and subsequently triggering the expression of HSPs [49]. HSF1 and HSF3 are thought to be key HSF genes in response to thermal stress in chicken. HSF1 is activated under low temperatures, while HSF3 is still activated at elevated temperatures and under long-term exposure [50].

In normal conditions, HSFs are maintained in a monomeric and inactive form and are typically linked to pre-existing HSPs in the cytoplasm. However, upon recognition of heat-stress signals, HSPs are released from HSFs, translocate to the nucleus, and transactivate the expression of a dozen cytoprotective genes that promote cell survival [7,51]. However, it was suggested that some other regulatory mechanisms exist besides HSFs [52].

All species have HSP genes varying in the pattern of their expression and functions [45]. For example, avian HSP90β mRNA is not induced by heat stress, such as HSP90α and β mRNA in rats and humans. Furthermore, the expression of HSP90β is not elevated by mitogenic inducers, such as serum or insulin, which induce HSP90α mRNA [53]. HSP70 is the most conservative and most common member of the HSP family in chicken and most other organisms. HSP70 plays a major role in the adaptive response to thermal stress in broiler chickens by improving antioxidant status, preventing lipid peroxidation, and increasing digestive enzyme activity [41]. However, the expression of HSP70 in commercial chicken in response to heat stress was found to be higher compared to that in native chicken. Similarly, the expression of HSPs in chicken differs from one tissue to another depending on the tissue’s susceptibility to heat stress [46]. Cedraz et al., on the other hand, reported that HSP70 expression was higher in local breeds under high-temperature conditions compared to that in a commercial line [50]. The overexpression of HSP90 also protects against several stimuli [54]. Furthermore, other families of HSPs, including sHSPs, are also involved in cell protection against environmental stresses. However, the protective roles of most HSPs were not extensively studied in poultry [11], and many more studies are needed to clarify how HSPs act in vivo. Studying the association between HSP expression during heat stress, and different physiological process-related chicken production and behavior helps improve their performance and welfare under such conditions.

## 4. HSPs Regulate Apoptotic and Antiapoptotic Signaling Pathways

Necrosis and apoptosis are two distinct routes of cell death. Necrosis is a passive form of cell death that is characterized by the loss of cell membrane integrity and the subsequent loss of its contents. Apoptosis is a highly regulated form of cellular autodestruction involving the orderly detachment of the cell, cytoplasm condensation, nuclear degradation, the fragmentation of the structure and function of cell components, and cell shrinkage; lastly, the cell becomes converted into an “apoptotic body” [48,55]. While cell survival responses to apoptosis processes in mammals are relatively well-described, chicken data are more infrequent and less informative. Apoptosis is coordinated by caspases, a family of cysteine proteases, which dismantle the cell and cleave intracellular proteins [56]. Caspase 6 (CASP6) is a caspase family member that has a central role in apoptosis. In the latest studies, CASP6 was downregulated in broiler chickens exposed to acute heat stress (24 h); however, CASP6 was upregulated after 12 days of exposure to chronic heat stress [57]. These authors suggested a correlation in enhancing apoptosis between ROS induced by heat stress and this high expression of CASP6 [57].

Mitochondria are the central mediator of apoptosis because they are implicated in the activation of the caspase via the release of cytochrome c, the most critical mediator of apoptosis [48,58]. Furthermore, the mitochondrial pathway is probably implicated in apoptosis induced by thermal stress by sensing oxidative stress and integrating and transducing stress signals. The crucial role of cytochrome c is combining with apoptotic protease activating factor-1 (Apaf-1) to contribute to the activation of caspase 9; therefore, apoptosis activation is well-known [59,60]. In cell cytosol, cytochrome c combines with Apaf-1, dATP, and procaspase-9 to induce active caspase-9 [61], which subsequently cleaves and activates executioner caspases [62]. The induction of HSPs following environmental stresses can prevent intrinsic (mitochondria and endoplasmic reticulum) and extrinsic apoptotic pathways [63]. In particular, HSP70, HSP90, and HSP27 play a crucial role in preventing apoptotic progression induced by various environmental stresses [63]. Apoptosome assembly is also a central step where HSPs can prevent apoptosome formation by selectively binding with Apaf-1 [58,64] and/or by their interaction with and inhibition of cytochrome c [65]. Using a cell-free system, HSP70 inhibits cytochrome c/dATP-mediated caspase activation but allows for the formation of Apaf-1 oligomers. HSP70 directly and selectively binds to the apoptosome’s central component (Apaf-1), but not to other apoptosome components, and suppresses the recruitment of caspases to the apoptosome complex [58]. Accordingly, the capacity of HSP70 to bind to specific proteins such as Apaf-1 raises its potential to inhibit caspase activation and suppress apoptosis [58]. This interaction of apoptosis and cellular-stress responses probably stabilizes cell survival in response to injury.

Members of the Bcl-2 family also play essential roles in apoptosis regulation, and Bax is a proapoptotic member of this family [66]. The overexpression of Bax is implicated in the loss and collapse of mitochondrial membrane potential, which leads to altered mitochondrial permeability [67], Bax translocation into mitochondria [66], and the subsequent release of mitochondrial cytochrome c [68]. There is evidence that HSP70 could block apoptosis by preserving mitochondrial membrane integrity by preventing BAX mitochondrial translocation and activation [69,70,71].

Samali et al. suggested that the lack of HSP induction is the cause rather than a consequence of cell death [48]. Moreover, the accumulation of HSPs was compatible with resistance to apoptosis in cells exposed to elevated temperatures (42 °C) [48]. These results indicated that a cell’s capability to produce HSPs at high-temperature conditions enables it to withstand this heat stress [48].

Phosphatidylinositol 3-kinase (PI3K)/Akt signaling mediates cell proliferation and survival as a classical pathway. Evidence from several experimental studies established that HSPs activate PI3K/AKT signaling [72]. The PI3K/AKT pathway prevents cellular damage and enhances cell survival by blocking apoptosis induced by several stressors in various models [73,74,75]. For instance, the induction of apoptosis in intestinal epithelial cells is widespread during heat stress and leads to impaired growth performance. However, cellular defense mechanisms in response to heat stress include the induction of HSPs, and prosurvival factors can alleviate the undesirable effects of heat stress. Among these factors, Akt functions as an essential part of suppressing heat-induced apoptosis. Thus, its activation enhances cellular survival via various mechanisms [63]. PI3K/Akt signaling contributes to the protection mechanism against the apoptotic process caused by thermal stress. Additionally, evidence from a trial on the IEC-6 cell line showed that exposure to thermal stress significantly increased Akt phosphorylation levels only at 2 h following treatment, indicating the early induction of PI3K/Akt signal [76]. 

Heat stress activates Akt through PI3K-dependent and -independent mechanisms, whereas oxidative stress-induced Akt activation is entirely regulated by PI3K, suggesting that the two stressors induce Akt activation by various pathways [77,78]. Cochaperone CDC37 enhances the association between HSP90 and the protein kinase, including Akt, to maintain their stability, activity, and signaling functions. The inhibition of Akt-HSP90 binding triggers Akt dephosphorylation and inactivation, leading to enhanced cell susceptibility to stimuli that induce apoptosis. The HSP90–Cdc37 complex promotes Akt stability by inhibiting its dephosphorylation by protein phosphatase 2A (PP2A) [79,80]. Moreover, HSP90 [81] and HSP27 [82] can regulate Akt activity in the apoptotic cascade. The interaction between HSP27 and AKT can form a complex with p38 mitogen-activated protein kinase (MAPK), which mediates the phosphorylation of HSP27 that binds to AKT and works as a scaffold protein to allow for the phosphorylation of AKT by PI3K to protect the cell from damage and apoptosis [82]. Taken together, these observations provide strong validation for HSP’s central role in the activation of Akt following heat or other types of stress. Furthermore, HSPs may also contribute to protecting Akt from proteasome-mediated degradation [83,84].

sHSPs, the first line of cellular defense against unfolding stress, prevent misfolding proteins via binding to early unfolding intermediates and sequestering them into sHSP/substrate complexes, leading to cellular protection and efficient refolding by the HSP70/Hsp100-based disaggregation system [85]. αB-Crystallin (CRYAB), a small heat-shock protein, protects the heart and myocardial cells against various cellular stresses, including heat stress. The overexpression of CRYAB appears to increase survival in heat-stressed myocardial cells in vitro and in vivo [11,86] through different pathways, including a reduction in F-actin aggregation (thus stabilizing the cytoskeleton), the regulation of the cell cycle, and the prevention of caspase-mediated apoptosis [87]. Increased Hsp20 expression showed significant protection against cardiomyocyte apoptosis and animal mortality [88]. The above authors suggested that phosphorylated Akt appears to be involved in the protective effect of Hsp20. Moreover, the pharmacological induction of HSPs, including Hsp70 and Hsp27, is involved in protecting against cardiomyocyte damage and apoptosis induced by heat stress [89,90,91]. Reduced levels of HSP70 and HSP90 during thermal stress were found to be associated with the formation of a myocardial lesion in broiler chickens, and the localization of HSP70 in the heart and blood vessels developed the protective mechanisms [32]. Together, HSPs and prosurvival factors are involved in the response to thermal stress and appear to be associated with alleviating cell damage and apoptosis under heat-stress conditions. Further studies are required for understanding the extent of crosstalk between HSP functions and other signaling networks involved in apoptosis pathways, and the physiological factors that modulate these signaling networks.

## 5. HSPs Mediate Thermotolerance

In poultry research related to environmental stressors, stress responses through HSP expression seem to be an essential biomarker to estimate the mechanism used for maintaining cellular integrity against stress damage. HSPs have a crucial role in regulating cellular stress resistance and stimulating thermotolerance following exposure to thermal stress [92,93]. Two mechanisms can confer thermotolerance and adaptation to high environmental temperatures: (1) the heat-shock response, a molecular defense system that emerged via the synthesis and expression of different HSP families and is characterized by short-acting and rapid expression following exposure to short-term sublethal heat stress; (2) heat acclimation, a mechanism that is activated as a response to long-term exposure to heat stress (chronic heat stress) and is characterized by long-acting effects [94,95].

There is a correlation between thermotolerance and the synthesis and accumulation of HSPs in the cell [48]. The protective role of HSPs is characterized by their capability to provide a state of thermotolerance, whereby moderate thermal-shock treatment renders cells adaptive and more resistant to subsequent and higher temperatures [58]. Following short exposure to thermal manipulation, several species have improved heat tolerances. This acquired thermal tolerance allows for adaptation and, subsequently, survival at extreme environmental temperatures [96]. Evidence from various studies indicates the crucial role of HSPs in enhancing thermotolerance in different species. The HSP expression of different species during heat stress is correlated with the level of stress that they naturally undergo [45]. For instance, the range of HSP overexpression can be greater than 1000-fold in *Drosophila* [97]. This vital role of HSPs includes immediate, rapid, and high expression in different cell types following heat stress [98], leading to the promotion of stress tolerance and consequently improving the survival rate of stressed cells [54].

In this sense, thermotolerance due to the expression of HSPs during heat stress prevents the aggregation or improper folding of proteins, maintains cellular homeostasis, and subsequently prevents cell injury [99]. The finding that HSPs act as molecular chaperones has contributed to the hypothesis that thermotolerance involves an enhanced capacity to refold heat-denatured proteins in a cell, which is associated with the increased expression of HSPs [100]. During hyperthermia, intracellular HSPs are considered to restrict protein unfolding and refold protein aggregates during heat-shock recovery. There is substantial support for this hypothesis, which considers that a subset of thermally sensitive proteins may provide heat tolerance to the cell. Different HSPs seem to chaperone different protein subsets [6,101].

Among all HSP families, those with a molecular size of about 70 kDa seem to be the most associated with thermal tolerance [102,103]. In vertebrates, HSP70 is always the first among HSP family members to reach its peak accumulation in different cell types subjected to a range of heat treatments, which supports the strong correlation between the development of thermotolerance and HSP expression [104,105,106,107]. Interestingly, in *Drosophila*, HSP70 has a crucial function in both thermotolerance and heat-shock regulation. Overexpressed HSP70 affects thermotolerance more than it does regulation; underexpressed HSP70 affects regulation more than it does thermotolerance. Furthermore, carboxyl-terminal deletion of HSP70 had a severe dominant-adverse effect on thermotolerance, but only minor effects on regulation; amino-terminal deletion strongly affects regulation but not thermotolerance [37]. 

The role of HSPs in protecting cells against heat stress is well-known. However, the specific function in thermotolerance acquisition in heat-manipulated chicks remains poorly understood, and different studies’ findings are inconsistent. An early study reported that early-age thermal manipulation by exposure of 5-day-old broilers to cyclic high temperatures (36 °C for 24 h) induced a long-term mechanism that acted to reduce hyperthermia during thermal stress, and the acquisition of improved thermotolerance was not associated with improved HSP response [108]. However, many different studies established a convincing argument suggesting the function of HSPs in thermotolerance development [103,105,109,110,111]. These conflicting findings propose that not all HSPs are involved in the development of thermotolerance acquisition; however, this development may be attributed to the nature of the cell model, or examined tissue, animal species, experiment conditions, or the abundance or deficiency of constitutive and inducible HSPs [103,112].

In both avian species and mammals, body temperature is neuroanatomically regulated by the preoptic anterior hypothalamus. Exposure to thermal conditioning before the maturation of the thermoregulatory system induces plastic alteration in the properties between warm- and cold-sensitive cells, leading to the modulation of thermotolerance [113]. In vital developmental stages, such as during the embryogenesis or early posthatch period, epigenetic adaptation modulates gene expression. Therefore, thermal manipulation in these early periods has an influential role in the modulation of epigenetic response. The exposure of cells or organisms to sublethal temperatures enhances cellular stress responses and develops a state of thermotolerance [48]. The thermal manipulation of chicken eggs by subjecting them to moderate thermal stress during susceptible periods of embryonic development was shown to improve the acquisition of thermotolerance during higher ambient temperatures during the posthatch period via inducing the expression of HSP70 [107,114,115], HSP90, HSP60, HSF-1 [116], HSF-3 [107,114], and HSF-4 [114] mRNAs in different tissue types (e.g., muscle, heart, liver, and brain) of broiler chickens several weeks after the termination of thermal manipulation. However, the expression values of these HSPs differ in a tissue-dependent manner during normal and heat-stress treatments [107,117]. In thermally manipulated broiler chicks, mRNA expressions of HSP70 showed a higher level in muscle than its expression in the brain and heart [107]. Several studies also supported these findings and reported that long-term enhanced HSP70 expression appears to be associated with enhanced thermotolerance acquisition in thermally manipulated chicks [114,115,116,118,119]. 

In the same manner, the sHSP family may also confer the protection of different tissue types as a response to thermal manipulation [120]. The overexpression of HSP27 and its murine homolog Hsp25 in CCL39 cells (derived from Chinese hamsters) enhanced thermotolerance and promoted actin filament stability under heat-stress conditions [121]. The thermal manipulation of broiler chicks during embryogenesis showed an increase in sHSP expression in the brain, suggesting that sHSPs are involved in thermotolerance in broiler chickens [120]. After 3 h of heat challenge at 37.5 °C, corticosterone (stress hormone) levels in thermally manipulated chicks were similar to those in the control group. Simultaneously, plasma corticosterone level was significantly higher in nonmanipulated chicks than that in manipulated chicks and/or control groups [113]. In some local chicken breeds that are naturally exposed to high temperatures, HSP70 and HSP90 are more highly expressed during heat stress than in commercial broiler lines [50], indicating the vital role of HSPs in adaptation and thermotolerance mediation.

Ultimately and most convincingly, the overexpression of HSPs is associated with the enhancement of thermotolerance in a wide range of cells and species during heat-stress conditions [37]. Furthermore, thermotolerance acquisition can be achieved by cell exposure to thermal manipulation, which enhances the rapid overexpression of HSPs, in particular, HSP70 and HSP27, via their roles in suppressing apoptosis in response to a wide range of stimuli [10,58,122].

Labunskay et al. [113] studied thermal manipulation in chicks to identify the molecular signals that regulate thermal tolerance. They found that 15 genes were induced during thermal manipulation. Most of these genes are associated with growth-related pathways, particularly with neuronal plasticity [113]. Among these identified genes, Ras GTP-binding proteins were reported to play a crucial role in regulating HSP70 induction following environmental stress [123]. HSP-mediated thermotolerance is likely essential for chicken’ cellular survival and adaptation during heat-stress conditions. Furthermore, cellular responses to thermal manipulation can confer thermotolerance and increase the potential to cope with heat stress via the functional roles of HSPs. Further studies are required to elucidate the ideal time of thermal manipulation that can provide optimal thermotolerance under heat-stress conditions, particularly near the market age.

## 6. HSPs Modulate Cellular Redox Homeostasis under Heat-Stress Conditions

Genetic selection for rapid growth rates makes broiler chickens more susceptible to oxidative damage. The exposure of broiler chickens to high ambient temperatures also causes oxidative stress, with both situations leading to biological damage, multiple pathological disorders, and impaired growth performance. Heat stress is a major source of oxidative damage in commercial strains of meat-and egg-type chicken by altering the oxidative status in several tissue types, leading to impaired metabolic function [124] and postmortem meat characteristics [125,126]. Acute [25,27] and chronic [27] heat stress associated with oxidative stress generation in the skeletal muscle of meat-type chicken significantly impairs growth and body weight. Oxidative stress also alters protein function and promotes susceptibility to proteolysis [127]. Broiler chickens treated with both acute and chronic heat stress showed a significant increase in protein oxidation in the pectoralis major muscle [27]. In particular, methionine and cysteine residues appeared to be less resilient to oxidation [128]. The oxidation of methionine reflects protein conformational alterations, protein unfolding, and degradation [129], while chicken exposed to acute heat stress only showed increased protein carbonyl levels in the liver [27]. Furthermore, the oxidation of muscle proteins may lead to protein proteolysis, affecting growth and body-weight gain. This impairment due to oxidative stress can negatively influence poultry immunity and cause severe poultry-production losses.

Oxidative stress induced by heat stress may be notable in all parts of the body; however, functional mitochondria are the most affected [24]. Calcium dysregulation and intracellular Ca^2+^ overload are associated with excessive ROS intracellular accumulation induced by heat stress, which may induce cytotoxicity and enhance apoptosis via the mitochondrial pathways [28].

The pentose phosphate pathway (PPP) is an essential antioxidant cellular defense system [130]. PPP is stimulated in response to oxidative stress and plays an essential role under heat stress since it is the principal intracellular source of NADPH, which acts as a cofactor for GPx [57,131]. Glutathione (GSH) is oxidized into glutathione disulfide (GSSG) by GPx enzymes. GSSG can be reduced back to GSH by GR in the presence of NADPH [132]. Following the stimulation of intracellular calcium, NADPH oxidase5 (NOX_5_) is stimulated as a response to Ca^2+^ overload, and it generates a tremendous amount of superoxides [133]. The upregulated expression of superoxide dismutases (SODs) was observed after exposure to chronic heat stress [57] as one of the antioxidant cellular defenses against superoxide accumulation [134]. SOD converts superoxides as free radicals into a nonradical species of hydrogen peroxide. Hydrogen peroxide is then converted by catalase into water and oxygen, and water by GPx in the presence of NADPH as a cofactor [135]. In particular, HSP27 phosphorylation appears to have a crucial role in activating glucose 6-phosphate dehydrogenase (G6PD), the rate-limiting enzyme in the pentose phosphate pathway [131]. The activation of G6PD was associated with increased levels of GSH and protection against oxidative stress [136]. Therefore, activated PPP glucose flux may enhance glucose biosynthesis and consumption [137]. Oxidative stress leads to increased lactate concentration in breast muscle [26]. A significant increase in glucose consumption was found in both HCT116 and mouse embryonic fibroblast due to enhanced PPP. Hence, glucose flux through the PPP may reduce lactate generation [137]. Because glucose is the most stable monosaccharide and lowers susceptibility to oxidation, it was suggested as the primary energy carrier in animal species. Thus, increased blood glucose level observed in avian species in comparison with those of mammals could be ready to use under stress conditions that have high energetic demand, thereby helping to maintain cellular functions [138].

HSPs regulate cellular redox homeostasis via their antioxidant function. The functional interplay between HSP70 and protein substrates involved in redox homeostasis is also correlated with the protective function of HSP70, which leads to preventing oxidative stress [125]. Guo et al. first observed that HSP70 can increase the activity of glutathione peroxidase (GPx) and glutathione reductase (GR), and its overexpression provides a more reducing intracellular environment during different stresses, indicating new insights into the mechanisms of cytoprotection provoked by HSP70 [139]. A strong correlation was found between reactive oxygen metabolites induced by hyperthermia and HSP70 induction in broiler chickens subjected to cyclic heat stress. Dietary supplements with ascorbic acid as an antioxidant showed a great reduction in HSP70 gene expression [140].

Since the microfilament network is one of the first sites for oxidative damage, in various in vitro models, thermally induced oxidative stress could impair the cytoskeleton via enhancing excessive actin polymerization [141]. F-actin is the fundamental component of the microfilament. Similar to an essential scaffold protein in cells, F-actin plays a crucial role in cell homeostasis and affects the cytoskeletal structure and cell membrane function [142]. Evidence revealed that the expression of HSP27 is causally associated with the modulation of actin microfilament responses after oxidative stress. In this sense, oxidative stress induces the fragmentation of F-actin in CCL39 cells in Chinese hamsters. Simultaneously, the overexpression of wild-type HSP27 following exposure to oxidative stress conferred protection against actin fragmentation. Furthermore, cell survival was increased with an increased level of HSP27 [143].

Cardioprotection by sHSPs against oxidative stress has been studied. Specifically, HSP27 (the homolog of HSP25) was reported to have an essential role in cardiomyocyte protection during several stresses. Heat-stress-induced HSP27 and its phosphorylation by MAPKAP-2 prevent cardiac H9c2 cell damage because phosphorylated HSP27 works as an endogenous antioxidant against the DOX-derived oxidants, such as oxygen free radicals and H_2_O_2_ [144]. Moreover, the pentose phosphate pathway is activated during cerebral ischemia and reperfusion. HSP27 is known for its dynamic phosphorylation during different stresses (e.g., oxidative, chemical, and heat stress). HSP27 phosphorylation could be a potential therapeutic target for treating ischemic stroke via its role in the activation of G6PD [131]. Increased oxidative stress leads to cardiomyocyte damage, but the overexpression of HSP20 was also found to alleviate doxorubicin-induced cardiac oxidative damage via the phosphorylation of Akt and BAD and to suppress the activation of caspase-3 [88].

## 7. Nutritional and Pharmacological Tuning of HSPs in Chicken

Several methods and intervention strategies were developed to induce HSP expression during the early stages of heat stress to prevent damage from extreme stress. Some plant-derived and pharmacological compounds with biological activities lead to the peak induction of HSPs at the time of stress with sustained induction for the first few days postinsult, leading to rapid cytoprotection, while other compounds attenuate the heat-stress-induced overexpression of HSPs through their biological activities, such as antioxidant and anti-inflammatory activities (Table 1).

Pretreatment with aspirin can increase HSP27 and CRYAB expression 45.62- and 6-fold, respectively, in the myocardium of heat-stressed chicken compared to the control group, indicating an adequate level to protect chicken myocardial cells against acute heat stress in an extracorporeal model [11,13]. The authors above suggested that myocardial cell damage could be prevented with overexpressed sHSPs, particularly HSP27 and CRYAB. Treatment with aspirin also showed a significant in vitro increase in the survival of chicken primary myocardial cells [11]. Pretreatment with coenzyme Q10 significantly increased the expression of HSP70 and accelerated its translocation into the nucleus, preventing the damage and apoptosis of chicken myocardial cells subjected to in vitro heat stress [91].

Medicinal plants and natural polyphenolic compounds have a potential effect in the alleviation of heat stress and oxidative damage in the immune organs of heat-stressed chicken [12,14] by modulating the expression of HSP27, HSP70, and HSP90 in the breast muscle, thymus, spleen, and bursa of Fabricius [145,146,147]. Furthermore, a significant increase in growth performance was found due to feeding a diet supplemented with resveratrol [145], indicating that resveratrol may modulate the expression of HSPs and consequently provide the protection of intestinal integrity and skeletal muscle under heat-stress conditions. Dietary resveratrol at the level of 400 mg·kg^−1^ decreased the expression of HSP70, HSP90, and NF-κB in the intestine of heat-stressed chicken after 2 weeks of treatment [146]. These findings indicated that resveratrol enhances intestinal morphology and mitigates intestinal mucosa damage under heat-stress conditions by regulating the mRNA and protein expression of HSPs. Decreased expression of HSP70 was observed in 35-day-old white Leghorn chicken that were fed dietary ascorbic acid under heat stress, indicating less of a stress response [17]. In another study conducted by Khan et al. [148], they reported that feeding broiler chickens with selenium-enriched probiotics under heat stress led to a significant increase in selenoproteins and downregulation in the expression of HSP60, HSP70, and HSP90. The downregulation of HSPs may be attributed to improved antioxidant capacity and alleviated heat stress due to dietary supplementation [148].

Supplementation with rosemary extract was also found to alleviate thermal stress in broiler chickens via inducing HSP expression and improving the antioxidant capacity [149]. Antioxidant supplementation in chicken feed/drinking water during heat stress could be used to mitigate the undesirable effects of heat stress through inducing HSP expression in breast muscle and intestinal segments [150,151]. Amino acids with antioxidant properties such as glutamine and taurine have been reported to modulate stress-induced HSP expression and improve cell survival against a variety of stressful stimuli. It can reduce lipid peroxidation in the breast muscle of broiler chickens reared under heat-stress conditions via modulating the expression of HSP70 and antioxidant related genes [152,153]. Interestingly, emerging evidence suggests that supplementation with the natural bioactive compounds effectively alleviate the heat stress in chicken due to their multiple biological activities. Curcumin, oregano and grape-seed extract with their proven anti-inflammatory and antioxidant activities have been observed to have various therapeutic benefits in chicken under heat-stress conditions [154,155,156] (Table 1).

## 8. Cytoprotective Role of HSPs against Heat-Stress-Induced Impairment of Intestinal Integrity

The gastrointestinal (GI) tract is the main part of the body in contact with the external environment. The intestinal epithelium plays an essential role in feed digestion, nutrient uptake, and protecting the body from pathogenic microbial infections. It acts as a natural barrier among the host and infection agent populations and the mucosal immune system. The gastrointestinal tract is highly sensitive to stressors such as oxidative stress induced by high environmental temperatures [41], which may cause changes in the intestinal mucosa, damage intestinal integrity, and increase intestinal permeability [29,157]. Damage of intestinal epithelial barrier caused by environmental stress may facilitate the translocation of pathogenic bacteria [157] and increase the absorption of toxic agents [141] into the body. Moreover, previous findings reported an association between chronic gut inflammation and the excessive production of reactive oxygen metabolites, leading to mucosal-barrier hyperpermeability [141,158,159]. Reactive oxygen metabolites can damage mucosal barrier integrity via the perturbation of the cytoskeletal network, which plays a central role in the stability of mucosal-barrier integrity [141]. HSP70 plays a crucial role in the alleviation of the deleterious effects of oxidative stress induced by heat stress. It has been reported that HSP70 can confer adequate protection of the intestinal mucosa against oxidative stress induced by acute heat stress by enhancing the antioxidant defense system and preventing lipid peroxidation [41]. Several investigations demonstrated that heat stress alters intestinal permeability by disturbing tight-junction (TJ) proteins [160]. Under stress, TJ changes in the intestines may be due to an increase in the production of free radicals causing alterations in protein kinase activities and changes in the phosphorylated status of TJ proteins [161]. Several studies indicated that, during stress, there is a rearrangement of zonula occludin, claudins, and occludins, which are TJ proteins controlling the integrity of the intestinal barrier [162]. HSPs play a vital role in preventing heat-induced intestinal-barrier destruction by encouraging the upregulation of occludin protein expression [163]. Hao et al. demonstrated that the overexpression of HSP70 might improve intestinal alkaline phosphatase activity, which indicates intestinal maturation and intestinal-barrier homeostasis [41].

TJ formation and enhancement are the priority of the function of occludin protein expression. The expression level of occluding proteins was shown to enhance cell-to-cell adhesion, while the peptide inhibitor of occludin prevents this adhesion in transfected fibroblast cells [164]. The treatment of filter-grown Caco-2 monolayers with high temperatures led to an increase in HSP expression. Furthermore, the suppression of HSP expression inhibits the upregulation of occludin protein expression in Caco-2 monolayers, and triggers obvious disturbance in the junctional localization of occludin proteins [163].

Some reports indicated that HSP70 is associated with stabilizing the actin cytoskeleton of intestinal cells, blocking their accumulation under environmental stress [165]. Furthermore, high levels of HSF1 and HSP70 are necessary for regulating actin fiber expression in gastrointestinal epithelial cells. 

Upon HSP activation, HSF1 is bound to the occludin promoter region by mediating expression upregulation and enhancing occludin involvement in junctional complexes [166]. HSP70 also preserves the GI epithelium, promotes the nutrient uptake and metabolic process, and exhibits an antioxidant effect [43]. HSP70 leads to induced cell proliferation and protein synthesis by attaching to recently synthesized polypeptides and mediating their folding; hence, it accelerates the healing of damaged tissue. HSP expression in the chicken gut may be an essential mechanism of antioxidant defense. Among HSP families, the functions of HSP70 in preventing intestinal integrity injury are the best-characterized. Hsp70 expression leads to a significant increase in antioxidant enzymes and inhibits the production of lipid peroxidation. It has been shown that upregulated HSP70/90 prevents heat-stress-induced intestinal mucosa damage by enhancing antioxidant status [43,167]. Hosseindoust et al. [168] reported that the level of HSP70 was significantly upregulated in different sections of the small intestine of chicken under acute heat stress. However, the expression of HSP family members in intestinal segments differs between genotypes under heat-stress conditions [39]. HSP70 preserves the intestinal mucosa from heat-stress-induced injuries by enhancing antioxidant capacity, preventing lipid peroxidation, minimizing cell damage, and controlling the production of lactic dehydrogenase (Figure 3) [43]. Collectively, these investigations propose that thermal stress induces damage to the chicken intestine and causes apoptosis to a greater degree. However, cell survival responses to heat stress include the induction of HSPs and other prosurvival factors, and the integration between these signaling pathways provides cytoprotection and inhibits heat-induced apoptosis.

## 9. Cytoprotective Role of HSPs against Heat-Stress-Induced Impairment of Immune Function

Thermal stress leads to immune dysfunction and apoptosis [169,170,171]. The levels of HSP27 and HSP70 were highly induced in immune organs under heat-stress conditions, while the use of antioxidants and immune regulators could decrease their levels to the normal range [169]. The weight of immune organs, phagocytosis capacity, macrophage incidence, and primary and secondary antibody responses were found to be decreased in broiler chickens following exposure to heat stress. Heat stress downregulated mitochondrial fission- and fusion-related genes and caused severe mitochondrial damage in the chicken spleen [171]. It also regulates the expression of proinflammatory cytokines in chicken lymphocytes. Innate immune cells generate cytokines and chemokines during heat-stress conditions [172]. Exposure to heat stress increased the protein levels of IL-1β and TNF-α and decreased those of IL-2 and INF-γ in the chicken spleen [171]. HSPs can modulate the immune and inflammatory responses. Members of HSP families are involved in both pro- and anti-inflammatory responses. The elevated expression of intracellular HSPs was found to improve cellular tolerance to inflammatory cytokines produced under heat-stress conditions. The upregulation of HSP70 in response to stress conditions contributes to inhibiting proinflammatory cytokine expression by preventing the transfer of nuclear factor-κB (NF-κB) to the nucleus [172]. These observations indicate that the elevated expression of HSPs during heat-stress conditions may confer cytoprotection by preventing cytokine production and improving cell tolerance.

Pathogenic species have a certain retained molecular structure known as pathogen-associated molecular patterns (PAMPs) that could be identified by a wide class of receptors present on the surface or within host cells (PRRs). PAMPs interact with PRRs and play an early role in the trigger of immunity. Heat stress negatively affects immunity and modulates immune responses to PAMPs. HSPs can be actively secreted by tumor or necrotic cells extracellularly and can present as substitute ligands for PRRs, hence triggering innate immune responses [173]. HSP60/70/90 plays a key role in the stimulation of innate immunity. HSP60 and HSP70 enhance avian macrophage cells for the secretion of proinflammatory cytokines (e.g., IL-1β, IL-6, IL-10, and IL-12).

Furthermore, HSP70 is involved in the activation and maturation of dendritic cells, and the modulation of their survival. However, the effect of HSPs on different immune cells appears to depend on their expression levels [174]. HSP70 also modulates adaptive immunity by providing the peptide fragments from stressed cells to a cytotoxic T cell [175]. Liu et al. reported that the expression of HSP27, HSP70, and HSP90 mRNAs in the bursa of Fabricius and spleen of 42-day-old chicken was upregulated due to exposure to high temperatures (37 ± 2 °C) for 15 days [145]. 

## 10. Cytoprotective Role of HSPs against Heat-Stress-Induced Impairment of Skeletal and Cardiac Muscles

Skeletal muscles consist of two main muscle fiber types depending on their mitochondrial content: slow-twitch (oxidative (Type I)) and fast-twitch (glycolytic (Type II)) muscle fibers [176]. Mitochondria are considered ROS generators in most cells. Oxidative muscles are vulnerable to ROS attacks because of their high mitochondrial activity; thus, they have greater antioxidant enzyme activity [177]. Heat stress induces oxidative damage to skeletal muscle tissue in chicken [27]. This involves surplus mitochondrial ROS generation due to the downregulation of avian uncoupling protein (avUCP) [178]. Skeletal muscle is somewhat unique concerning apoptosis. In preserving cellular homeostasis, HSP70 plays a universal function in many cellular aspects, such as cellular stress defense, apoptosis, cellular tolerance, and energy metabolism. HSP70 maintains cell survival and prevents the initiation of apoptosis in skeletal muscle when the cell is exposed to chronic stimulation and loading [93]. Elevated-heat-denatured proteins have detrimental effects on cells. However, HSP70 can confer cellular protection by trapping and refolding denatured proteins. Moreover, the phosphorylation of the sHSP family may also contribute to cytoskeletal protection [179] and assist in cytoskeletal stabilization or repair [179,180,181].

In broilers, acute heat treatment elevated HSFs and HSPs compared to chronic heat-stress treatment [52]. HSF2 and HSF3 are abundant in muscle and alterations in the expression of HSF genes were not marked in the muscle after heat treatments. 

The exposure of skeletal muscle to heat stress increases the expression of HSP70; hence, muscle mass can be protected and maintained, indicating that elevated HSP70 expression may be involved in reducing heat-stress-induced oxidative stress in skeletal muscle [182]. Liu et al. [183] suggested that the cytoprotection effect of HSP70 may be due to its ability to modulate pro- and anti-inflammatory cytokines. As mentioned above, heat stress leads to calcium dysregulation and intracellular Ca^2+^ overload, which results in cytotoxicity and apoptosis via mitochondrial pathways. Heat stress can impair membrane lipids and modulate the activity of sarcoendoplasmic reticulum Ca^2+^-ATPase (SERCA, the key protein for intracellular Ca^2+^ elimination, and for thermotolerance and muscular function), leading to the disruption of cell function and cellular metabolism. HSP70 interacts with SERCA, prevents its thermal inactivation, and protects its function by stabilizing, indicating the association between upregulated HSP70 with decreased muscle degeneration [184,185]. This reparation of SERCA ATPase inhibits oxidative stress-related muscle impairment [186] (Figure 4).

The heart is the most important organ in birds and mammals, and the clinical studies of human cases showed that thermotolerance is reduced in patients with cardiovascular diseases. Additionally, the enzymes related to heart injury could be increased in the blood serum and supernatant of myocardial cells under heat-stress conditions. Furthermore, it appears that heat-induced sudden death is mainly associated with myocardial cell injury [86,87,88,90,91]. There is an association between the cytoprotective effects and HSP functions in cardiac tissue [90,91]. In a recent study by Srikanth et al. [187] to evaluate the heat-stress responses of lowland and highland chicken, results showed that early acclimation to heat stress leads to adequate responses to heat stress. HSPB7 showed significant upregulation in lowland chicken’ cardiac and skeletal muscles, while downregulated HSPB7 was found in highland chicken. Several apoptosis pathways were also activated in highland chicken, suggesting considerable cellular damage. These results indicated the effect of acclimation on counteracting heat-stress-induced protein aggregation in chicken muscle. In other words, skeletal muscle adapts to heat stress through the induction of HSPs. HSPs work to preserve cell homeostasis, promote repair from damage and provide cell tolerance against future stress. Accordingly, more research is required to determine how much this varies between breeds and target tissue types, and to what extent adaption can improve cellular responses to heat stress. The role of HSPs in modulating cell signaling pathways during adaption to heat stress also needs further investigation.

## 11. Conclusions

Heat stress plays a prominent role as one of the most critical environmental stress types affecting commercial poultry. Heat stress and the associated oxidative stress lead to cell dysfunction and apoptosis. Exposure to heat-stress conditions stimulates molecules ready to mediate cell death and survival signals and supports cell tolerance and/or damage recovery. Among these biomolecules are HSPs, which protect cells against both heat and oxidative stress through various functions, including the development of thermal endurance, the modulation of apoptotic signaling pathways, and the regulation of cellular oxidative states. Some feed additives contribute to controlling the rhythm of the gene expression of HSPs, either by rapid activation immediately after the occurrence of heat stress or by reducing this expression under the long-term exposure to heat stress as a result of the various biological effects of these molecules, such as improving oxidative status, anti-inflammatory activity, and thermotolerance.

Exposure of cells to relatively high temperatures during embryogenesis or during the posthatch period by thermal manipulation appears to stimulate cell survival responses that provide protection and improve thermotolerance and resistance to related future stress. The investigation of dietary manipulation to induce HSPs in susceptible tissue types, such as the intestine, skeletal muscle, and immune organs, under heat-stress conditions is essential for providing rapid protection and helpful knowledge on the mechanism of molecular activity that ensures optimal protection from thermal stress. In this work, we reviewed the functional roles of HSPs in cell survival and the physiological factors that modulate their expression under heat-stress conditions on the cellular and systemic levels. However, the mechanisms by which these molecules confer cytoprotection against heat stress still need further research to be elucidated. Further work is also required to understand the potential of using HSPs in a therapeutic setting to alleviate the undesirable effects of heat stress in commercial chicken.

## Figures and Tables

**Figure 1 animals-10-02407-f001:**
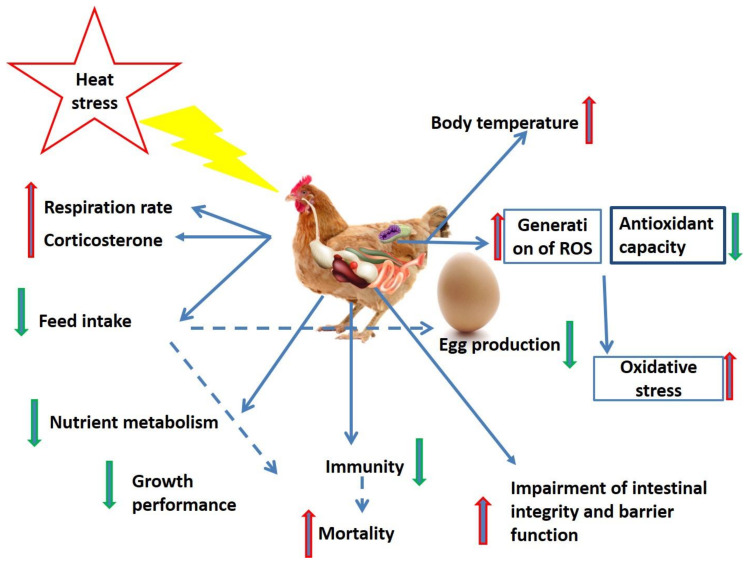
Undesirable effects of heat stress on commercial chicken.

**Figure 2 animals-10-02407-f002:**
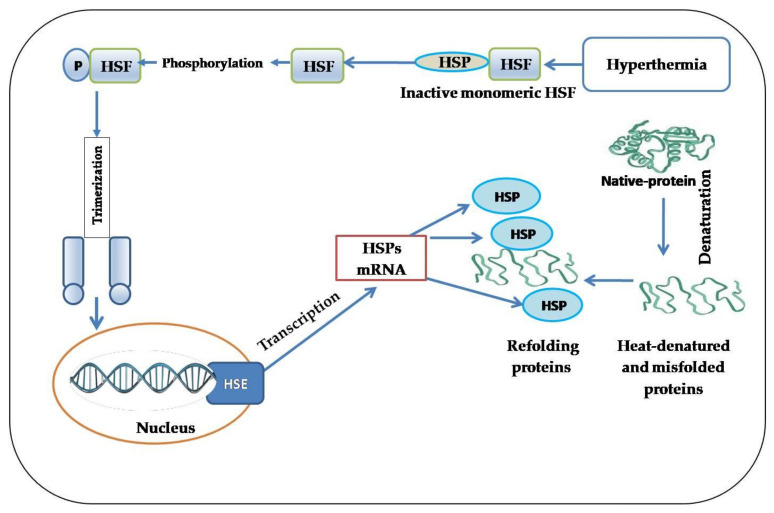
Activation and induction of heat-shock proteins in response to hyperthermia.

**Figure 3 animals-10-02407-f003:**
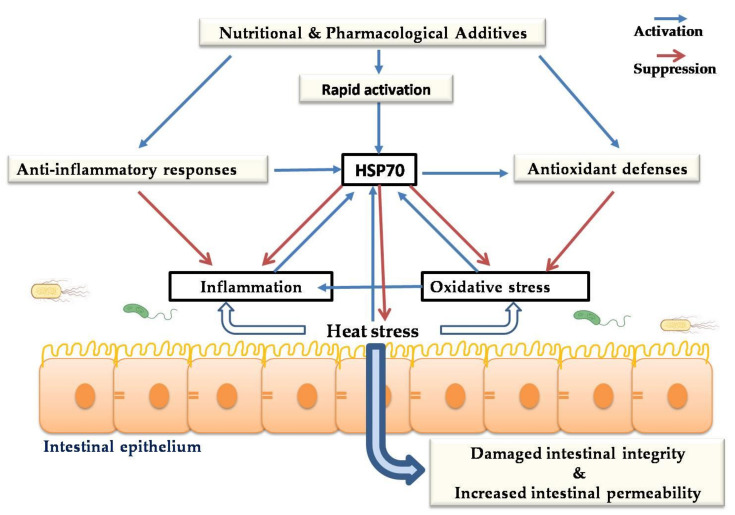
Potential mechanisms underlying the cytoprotective role of HSPs against heat-stress-induced impairment of intestinal integrity.

**Figure 4 animals-10-02407-f004:**
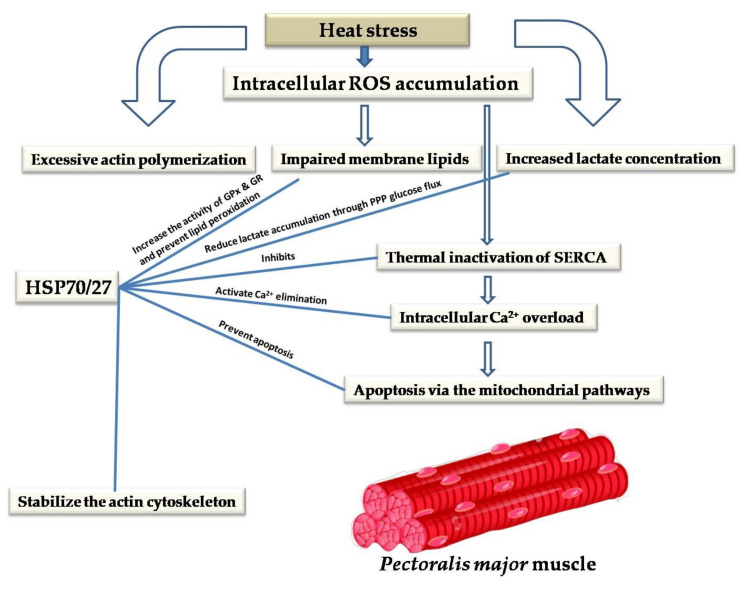
Potential mechanisms underlying the cytoprotective role of HSPs against heat-stress-induced impairment of skeletal muscle.

**Table 1 animals-10-02407-t001:** Effect of nutritional and pharmacological compounds on heat-shock proteins (HSPs) expression in heat-stressed chicken.

Bioactive Compounds and Medicinal Herbs *	Regulation of HSP Expression	Tissue	Reference
Flavonoids genistein and hesperidin	Decreased HSP70	Breast muscle	[14]
Ascorbic acid	Decreased HSP70	Heart	[17]
Resveratrol	Decreased HSP27, HSP70, and HSP90 by attenuating heat-stress-induced overexpression	Bursa of Fabricius and spleen	[145]
	Increased HSP27 and HSP90 by increasing heat-stress-induced low expression	Thymus	[145]
	Decreased HSP70 and HSP90 after 15 days of heat stress by attenuating heat-stress-induced overexpression	Jejunal villi	[146]
*Artemisia annua* L. (*A. annua*)	Decreased HSP70 and HSP90	Breast muscles	[147]
Rosemary	Increased HSP70 for 5–10 h; increased HSP27 at 3 h; increased CRYAB at 0 and 3–10 h	Heart	[149]
Vitamin E (VE) and selenium (Se)	Increased HSP90, HSP70, and HSP60 with Se and Se + VE supplementation. Increased HSP90 and HSP70 with VE supplementation.	Breast muscle	[150]
Zinc oxide nanoparticles	Increased HSP70 &HSP90	Jejunum, Duodenum, and ileum	[151]
Glutamine	Increased HSP70	Breast muscle	[152]
Taurine	Decreased HSP90	Breast muscle	[153]
Curcumin	Decreased HSP70 and HSP90	Breast muscles	[154]
Grape-seed extract (GSE) and vitamin C	Decreased HSP70	Heart	[155]
*Origanum compactum* and *Curcuma xanthorrhiza*	Decreased HSP70	Heart	[156]

* All bioactive compound and medicinal herbs were administered orally either with the diet or the drinking water. Details about the dosage and experimental heat-stress conditions are described in the corresponding reference.

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
