# Peer review of "Modulation of Heat-Shock Proteins Mediates Chicken Cell Survival against Thermal Stress"

_animals, 2020, doi:10.3390/ani10122407_

Round 1

Reviewer 1 Report

The manuscript titled “Modulation of heat shock proteins mediates chicken cell survival against thermal stress.” by  Shehata and others attempted to review a topic as stated in their title.

The manuscript is, however, largely inconsistent with the title and contains numerous scientific and writing problems.

A review paper should present a summary of current knowledge and problems and integrate them to present potential interpretations and future directions.

The current manuscript lacks any strategic structure, but just a list of items which are too general and no explicit link to the aim of the manuscript.

A large portion of the manuscript (which are titled 6, 8, 9, and 10) was used to describe “Cytoprotective effect”, but there is no definition of the Cytoprotective effect and those chapters have no explicit connections. Basically, half of the manuscript is organized with a groundless concept.  

The manuscript should be re-organized with a solid conceptual foundation and major problems and potential solutions to improve chicken growth and carcass quality under heat stress.    

Content under 7. Nutritional and pharmacological tuning of HSPs in chickens does not align with the title and most of the items listed in Table 1 are not discussed. This whole part is another example showing the lack of the strategic structure of the manuscript.

There are too many grammar problems to list. The authors should consult a professional English editor. There are several sentences that are very close to the original sentences in the referenced articles. These are at the borderline of plagiarism.

Figure qualities are unprofessional. Colors are unnecessary(Figure 3 & 4) and there are unnecessary trivial things (transcription & translation in Figure 2).     

The conclusion does not include any specific and solid knowledge because there were no major problems declared and the main body of the manuscript was filled with various items in a descriptive way.

Author Response

The manuscript titled “Modulation of heat shock proteins mediates chicken cell survival against thermal stress.” by Shehata and others attempted to review a topic as stated in their title.

The manuscript is, however, largely inconsistent with the title and contains numerous scientific and writing problems.

Response: We thank the reviewer for the thoughtful review. We have edited the manuscript following your comments and suggestions that greatly improved the quality of the manuscript. We also edited the language by the aid of MDPI English editing service to improve and correct the grammatical errors.

A review paper should present a summary of current knowledge and problems and integrate them to present potential interpretations and future directions.

The current manuscript lacks any strategic structure, but just a list of items which are too general and no explicit link to the aim of the manuscript.

Response: In this work, we have clearly described the problem (heat stress) in a separated section and aimed to discuss the different cell survival responses to heat stress via HSPs function (Thermotolerance, Apoptosis pathway, mediation of cellular redox homeostasis). Then we have discussed how to modulate  HSPs expression through feed additives in chicken. Then we have discussed some example for HSPs cytoprotective function and selected for this purpose the most important tissues (Intestine, immune cells and skeletal muscle) which are of interest of the majority of poultry researchers and directly affect the growth performance of chickens.

A large portion of the manuscript (which are titled 6, 8, 9, and 10) was used to describe “Cytoprotective effect”, but there is no definition of the Cytoprotective effect and those chapters have no explicit connections. Basically, half of the manuscript is organized with a groundless concept.  

Response: We have discussed the available data from chicken studies, however, it were quite limited.

The manuscript should be re-organized with a solid conceptual foundation and major problems and potential solutions to improve chicken growth and carcass quality under heat stress.    

Response: We have clearly described the problem and discussed the potential solution such as thermal manipulation, use of antioxidants and the use of feed additives.

Content under 7. Nutritional and pharmacological tuning of HSPs in chickens does not align with the title and most of the items listed in Table 1 are not discussed. This whole part is another example showing the lack of the strategic structure of the manuscript.

Response: In this section, we briefly have referred to a rapid expression of HSPs during the exposure to heat stress and attenuation effect by the long term of exposure to heat stress, via the several feed additives and the table content gives clear examples for these effects.

There are too many grammar problems to list. The authors should consult a professional English editor. There are several sentences that are very close to the original sentences in the referenced articles. These are at the borderline of plagiarism.

Response: Thanks a lot for this observation. The manuscript has been revised by MDPI English editing service. We attach the Certificate of Editing for your consideration.

Figure qualities are unprofessional. Colors are unnecessary (Figure 3 & 4) and there are unnecessary trivial things (transcription & translation in Figure 2).   

Response: Thanks a lot for this observation. We have improved the mentioned figures.  

The conclusion does not include any specific and solid knowledge because there were no major problems declared and the main body of the manuscript was filled with various items in a descriptive way.

Response: We thank the reviewer for this comment. We have improved the conclusion.

Reviewer 2 Report

The Authors have investigated an interesting topic and the theme has been properly described.

I would like to congratulate authors for the good-quality of the article, the literature reported used to write the paper, and for the clear and appropriate structure. The manuscript is well written, presented and discussed, and understandable to a specialist readership.

In general, the organization and the structure of the article are satisfactory and in agreement with the journal instructions for authors. The subject is adequate with the overall journal scope.

The work shows a conscientious study in which a very exhaustive discussion of the literature available has been carried out.

The introduction provides sufficient background, and the other sections include results clearly presented and analyzed exhaustively.

However, the paper needs some revisions before a possible acceptance:

Lines 37-46: Add a couple of reference to support the statements;

Lines 69, 445, 483, 551: Correct the references according to the journal's guidelines;

The Conclusion section could be further improved.

So, I recommend the acceptance of the paper in Animals after revision.

Author Response

The Authors have investigated an interesting topic and the theme has been properly described.

I would like to congratulate authors for the good-quality of the article, the literature reported used to write the paper, and for the clear and appropriate structure. The manuscript is well written, presented and discussed, and understandable to a specialist readership.

In general, the organization and the structure of the article are satisfactory and in agreement with the journal instructions for authors. The subject is adequate with the overall journal scope.

The work shows a conscientious study in which a very exhaustive discussion of the literature available has been carried out.

The introduction provides sufficient background, and the other sections include results clearly presented and analyzed exhaustively.

However, the paper needs some revisions before a possible acceptance:

Response: We thank the reviewer for the thoughtful review of our work and kind words. We have thoroughly re-reviewed the manuscript and corrected any errors accordingly.

Lines 37-46: Add a couple of reference to support the statements;

Response: Thanks for this observation. Done, L 41

Lines 69, 445, 483, 551: Correct the references according to the journal's guidelines;

Response: We thank the reviewer for this observation. Done as suggested. L 63, 446, 477, 564

The Conclusion section could be further improved.

Response: Thanks a lot for this comment. We have improved this part. Line 601-612

So, I recommend the acceptance of the paper in Animals after revision.

Response: We hope the manuscript now meets the quality of the publication. Thank you again for your careful review.

Reviewer 3 Report

Animals

Title: Modulation of heat shock proteins mediates chicken cell survival against thermal stress

Manuscript Number: animals-1012227

The manuscript reviewed modulation of heat shock proteins mediates chicken cell survival against thermal stress, and showed that oxidative stress, inflammation, and apoptosis were involved in abobe mechanism. The title and the contents meet the requirements of animals. However, However, some contents need to be revised before publishing the manuscript in "Animals" in order to improve the quality of the manuscript. For example, in the manuscript, there are many errors on singular and plural, and tense; and the format is not uniform. Some comments are as follows.

For Abstract

  1. L26-27, change “Heat shock proteins ” to “Heat shock proteins (HSPs)”.
  2. L30-31,  “potential of both HSP70 and HSP27 to serve as broad-spectrum inhibitors of apoptosis pathways”?

For 1. Introduction

  1. L44-46, The following sentence is incomplete, please rewrite it. “Among these mechanisms, promoting the expression of cell survival-related genes, leading to early activation of some molecules that work as molecular chaperones and contribute to cell survival process during heat stress such as heat shock factors (HSFs) and heat shock proteins (HSPs).” Correct similar mistakes please.
  2. L 54-55, change “Hsp70 and HSP27 have” to “HSP70 and HSP27 have”.
  3. L 56, change “Heat shock proteins include small heat shock protein” to “HSPs include small HSPs”.
  4. L 61-63, About “Recent developments in the field of biomedical research have led to a renewed interest in HSPs due to their vital roles not only in thermal stress but also in cell survival during exposure to other different stresses [7].” [7] was published in 1997, [7] did not support “Recent developments in the field of biomedical research ”.
  5. L 67-68, About “The last two decades have seen a growing trend of therapeutic targeting of HSPs via nutritional and pharmacological agents towards alleviating heat stress in commercial chickens [9–11]”, The references [9–11] are research articles and were published in 2013 and 2016, respectively. Thus the references [9–11] did not support above contents.
  6. L70, Chang “heat-stressed broiler” to “heat-stressed broilers [12]”.

For 2. Heat stress

  1. L 79, change “malondialdehyde (MDA) levels” to “malondialdehyde (MDA) level”.
  2. L 79-80, About “Chronic heat stress reduces the body gain [17], the carcass yields and immunological parameters [18], ”  “Chronic heat stress reduces immunological parameters” is not appropriate.
  3. L80-81 Change “the levels of liver HSP70 mRNA expression” to “the level of liver HSP70 mRNA expression”.
  4. L 82,  Is “failure of oxidation process” right?
  5. L 85-86, About “It has recently been”, the reference [20] was published in 2016, “recently” here is not appropriate. Please correct similar mistakes.
  6. L 86, Change “findings demonstrated that heat stress results” to “findings demonstrated that heat stress resulted”. Please correct similar mistakes.
  7. L 98-99, Can “It is also likely that heat stress causes a rise in ROS production, which induces deregulated intracellular Ca2 +concentrations”lead to“various other biological disturbances”?
  8. L108, The citation format of references is inconsistent. Such as “(Strong, 2014; Hirakawa et al., 2020)” here, “[26]” on line 96
  9. L108, Pleae rewrite “ High temperature leads to increased”
  10. L112-113, About “Recently, research has shown an association between severe heat stress-caused lesions and the expression of HSP70 in chickens [29,30]”, the referecne [29,30] were published in 2004 and 2008, respectively. Is “Recently” here appropriate? There were many similar mistakes in the manuscript.
  11. L 120, About “In previous studies [39–41]”. This is a review and not a research article. “previous studies” here is not appropriate.
  12. L 126, About “(Figure 1)”, The contents in front of Figure 1 does not support the contents of Figure 1.

For 3.A brief history of HSPs in chickens

  1. L140-142, About “HSPs can be categorized into several protein families dependent on their molecular mass, including the HSP100, HSP90, HSP70, HSP60, HSP40, and small heat shock proteins (sHSPs)”, Are “HSP100, HSP90, HSP70, HSP60, HSP40, and small heat shock proteins (sHSPs)” protein families?
  2. L 143-144, About “the most massive increase”, compared with what?
  3. L 146, “In the vertebrates,” and “A brief history of HSPs in chickens” are contradictory
  4. The contents of this part did not support the title of this part, Please rewrite the title and the contents.

For 4. HSPs regulate apoptotic and antiapoptotic signaling pathways

  1. There's no informationon HSPs in the first paragraph of this section.

For 5. Heat shock proteins mediate thermotolerance.

  1. L 250, change “Heat shock proteins mediate thermotolerance” to “ HSPs mediate thermotolerance”.

For 7. Nutritional and pharmacological tuning of HSPs in chickens.

  1. L 436-437, change “expression of HSP27, Hsp70, and HSP90” to “ expression of HSP27, HSP70, and HSP90”.
  2. L 437-443, “a significant increase in growth performance was found due to a feeding diet supplemented with resveratrol[151], indicating the resveratrol role of elevated expression of HSPs for intestinal integrity and skeletal muscle.” and “Dietary resveratrol at the level of 400mg. kg-1 decreased expression of HSP70, HSP90, and NF-κB in the intestine of heat-stressed chickens after two weeks of treatment [152]. ” are contradictory. Therefore, “These findings” can not “indicate that resveratrol enhances the intestinal morphology and mitigate intestinal mucosa damages under heat stress conditions by regulating the mRNA and protein expression of HSPs. ”

For 8. Cytoprotective role of HSPs against heat stress-induced impairment of intestinal integrity

  1. L 453, change “The main part of the body in contact with the external environment is the gastrointestinal (GI) tract. ” to “Gastrointestinal (GI) tract is The main part of the body in contact with the external environment”.
  2. L 477-478, About “As mentioned above, HSP70 leads to induced cell proliferation and protein synthesi ”, “above” did not “menteioned” that “HSP70 leads to induced cell proliferation and protein synthesi ”.

For 9. Cytoprotective role of HSP against heat stress-induced impairment of immune function

  1. L 493, change “Cytoprotective role of HSP against” to “Cytoprotective role of HSPs against”.

Author Response

The manuscript reviewed modulation of heat shock proteins mediates chicken cell survival against thermal stress, and showed that oxidative stress, inflammation, and apoptosis were involved in abobe mechanism. The title and the contents meet the requirements of animals. However, some contents need to be revised before publishing the manuscript in "Animals" in order to improve the quality of the manuscript. For example, in the manuscript, there are many errors on singular and plural, and tense; and the format is not uniform. Some comments are as follows.

Response: We thank the reviewer for the thoughtful and thorough review that greatly improved the quality of the work and make our review more balanced.

For Abstract

  1. L26-27, change “Heat shock proteins ” to “Heat shock proteins (HSPs)”.

Response: Thanks for this observation, done as suggested. L26-27

  1. L30-31,  “potential of both HSP70 and HSP27 to serve as broad-spectrum inhibitors of apoptosis pathways”?

Response: Thanks for this comment. We have rewritten the sentence. L30

For 1. Introduction

  1. L44-46, The following sentence is incomplete, please rewrite it. “Among these mechanisms, promoting the expression of cell survival-related genes, leading to early activation of some molecules that work as molecular chaperones and contribute to cell survival process during heat stress such as heat shock factors (HSFs) and heat shock proteins (HSPs).” Correct similar mistakes please.
  2. Response: We thank the reviewer for this comment. We haverewritten the sentence. L43-46
  3. L 54-55, change “Hsp70 and HSP27 have” to “HSP70 and HSP27 have”.

Response: Thank a lot for this observation. Done, L54

  1. L 56, change “Heat shock proteins include small heat shock protein” to “HSPs include small HSPs”.

Response:Thank a lot for this observation. Done, L56

  1. L 61-63, About “Recent developments in the field of biomedical research have led to a renewed interest in HSPs due to their vital roles not only in thermal stress but also in cell survival during exposure to other different stresses [7].” [7] was published in 1997, [7] did not support “Recent developments in the field of biomedical research ”.

Response: Thank a lot for this observation. We have updated the reference, L63

  1. L 67-68, About “The last two decades have seen a growing trend of therapeutic targeting of HSPs via nutritional and pharmacological agents towards alleviating heat stress in commercial chickens [9–11]”, The references [9–11] are research articles and were published in 2013 and 2016, respectively. Thus the references [9–11] did not support above contents.

Response: We thank the reviewer for this comment. We improved the sentence to be suitable with the references. L66

  1. L70, Chang “heat-stressed broiler” to “heat-stressed broilers [12]”.

Response: Thanks for this observation. Done as suggested, L 69

For 2. Heat stress

  1. L 79, change “malondialdehyde (MDA) levels” to “malondialdehyde (MDA) level”.

Response: Thanks for this observation. Done as suggested, L 78

  1. L 79-80, About “Chronic heat stress reduces the body gain [17], the carcass yields and immunological parameters [18], ”  “Chronic heat stress reduces immunological parameters” is not appropriate.

Response: Thanks a lot for this observation. We have improved it, L 79

  1. L80-81 Change “the levels of liver HSP70 mRNA expression” to “the level of liver HSP70 mRNA expression”.

Response: Thanks for this observation. Done as suggested, L 80

  1. L 82,  Is “failure of oxidation process” right?

Response: Thanks for this observation. We have removed this part.

  1. L 85-86, About “It has recently been”, the reference [20] was published in 2016, “recently” here is not appropriate. Please correct similar mistakes.

Response: Thanks a lot for this observation. Done as suggested,L97,98,466, 497

  1. L 86, Change “findings demonstrated that heat stress results” to “findings demonstrated that heat stress resulted”. Please correct similar mistakes.

Response: Thanks a lot for this observation.Done as suggested, L 85 and all similar mistakes have been also corrected along the manuscript.

  1. L 98-99, Can “It is also likely that heat stress causes a rise in ROS production, which induces deregulated intracellular Ca2 +concentrations” lead to“various other biological disturbances”?

Response: Thanks a lot for this observation. Done as suggested, L 99

  1. L108, The citation format of references is inconsistent. Such as “(Strong, 2014; Hirakawa et al., 2020)” here, “[26]” on line 96

Response:Thanks a lot for this observation. Done as suggested, L105-108

  1. L108, Please rewrite “ High temperature leads to increased”

Response: Thanks a lot for this observation. Done as suggested, L 104

  1. L112-113, About “Recently, research has shown an association between severe heat stress-caused lesions and the expression of HSP70 in chickens [29,30]”, the reference [29,30] were published in 2004 and 2008, respectively. Is “Recently” here appropriate? There were many similar mistakes in the manuscript.

Response: Thanks a lot for this observation. We have improved it, L 109

  1. L 120, About “In previous studies [39–41]”. This is a review and not a research article. “previous studies” here is not appropriate.

Response:Thanks a lot for this observation. We have rewritten the sentence, L 117-119

L 126, About “(Figure 1)”, The contents in front of Figure 1 does not support the contents of Figure 1.

Response: Thanks a lot for this observation. We have moved it, L 77

For 3. A brief history of HSPs in chickens

  1. L140-142, About “HSPs can be categorized into several protein families dependent on their molecular mass, including the HSP100, HSP90, HSP70, HSP60, HSP40, and small heat shock proteins (sHSPs)”, Are “HSP100, HSP90, HSP70, HSP60, HSP40, and small heat shock proteins (sHSPs)” protein families?

Response: Yes, they are protein families, however, we have removed “protein”, L 136

  1. L 143-144, About “the most massive increase”, compared with what?

Response: Thanks a lot for this observation.We have rewritten it, L 138

  1. L 146, “In the vertebrates,” and “A brief history of HSPs in chickens” are contradictory

Response: Thanks a lot for this observation. We have rewritten the title to be more general and due to limited data from chickens. L126

  1. The contents of this part did not support the title of this part, Please rewrite the title and the contents.

Response: Thanks a lot for this observation. We have rewritten the title to be more general and due to limited data from chicken studies.

For 4. HSPs regulate apoptotic and antiapoptotic signaling pathways

  1. There's no information on HSPs in the first paragraph of this section.

Response: Thanks a lot for this observation. Indeed, this part was an introduction and a definition.

For 5. Heat shock proteins mediate thermotolerance.

  1. L 250, change “Heat shock proteins mediate thermotolerance” to “ HSPs mediate thermotolerance”.

Response: We thank the reviewer for this observation. Done as suggested, L 257

For 7. Nutritional and pharmacological tuning of HSPs in chickens.

  1. L 436-437, change “expression of HSP27, Hsp70, and HSP90” to “ expression of HSP27, HSP70, and HSP90”.

Response: We thank the reviewer for this observation. Done as suggested, L 437

  1. L 437-443, “a significant increase in growth performance was found due to a feeding diet supplemented with resveratrol[151], indicating the resveratrol role of elevated expression of HSPs for intestinal integrity and skeletal muscle.” and “Dietary resveratrol at the level of 400mg. kg-1 decreased expression of HSP70, HSP90, and NF-κB in the intestine of heat-stressed chickens after two weeks of treatment [152]. ” are contradictory. Therefore, “These findings” can not “indicate that resveratrol enhances the intestinal morphology and mitigate intestinal mucosa damages under heat stress conditions by regulating the mRNA and protein expression of HSPs. ”

Response: Thanks for this comment. We have rewritten the sentence, L 439.However, A substance like resveratrol may induce HSP expression once the organism exposed to heat stress, anddecrease HSP expression after long-term of exposure as a result to its biological activity.

For 8. Cytoprotective role of HSPs against heat stress-induced impairment of intestinal integrity

  1. L 453, change “The main part of the body in contact with the external environment is the gastrointestinal (GI) tract. ” to “Gastrointestinal (GI) tract is The main part of the body in contact with the external environment”.

Response: We thank the reviewer for this observation. Done as suggested, L 453-454

  1. L 477-478, About “As mentioned above, HSP70 leads to induced cell proliferation and protein synthesis ”, “above” did not “mentioned” that “HSP70 leads to induced cell proliferation and protein synthesis ”.

Response: Thanks for this comment. We have rewritten the sentence, L 492-493

For 9. Cytoprotective role of HSP against heat stress-induced impairment of immune function

  1. L 493, change “Cytoprotective role of HSP against” to “Cytoprotective role of HSPs against”.

Response: Thanks a lot for this observation. Done as suggested, L512.

Round 2

Reviewer 1 Report

The revised manuscript titled "Modulation of Heat-Shock Proteins Mediates Chicken Cell Survival against Thermal Stress" by Shehata and others attempted to review a topic as stated in their title.

It has accommodated critiques and suggestions from the first round of review.

However, there remains one main problem. The reviewer disagrees with the authors’ view on the table 1. It is a two-page long, extensive table that contains various information of fifteen bioactive compounds. Only three compounds were covered in the text and the other twelve compounds are dangling. If these are trivial items that do not deserve mentioning in the text, then they should not be there. The reviewer suggests either delete the table and focus on the three compounds in the text or describe the other twelve compounds at least briefly and keep the table.

Author Response

The revised manuscript titled "Modulation of Heat-Shock Proteins Mediates Chicken Cell Survival against Thermal Stress" by Shehata and others attempted to review a topic as stated in their title.

It has accommodated critiques and suggestions from the first round of review.

However, there remains one main problem. The reviewer disagrees with the authors’ view on the table 1. It is a two-page long, extensive table that contains various information of fifteen bioactive compounds. Only three compounds were covered in the text and the other twelve compounds are dangling. If these are trivial items that do not deserve mentioning in the text, then they should not be there. The reviewer suggests either delete the table and focus on the three compounds in the text or describe the other twelve compounds at least briefly and keep the table.

Response: We would like to thank the reviewer for his thoughtful review and the great suggestion. All your comments have been taken into the consideration and done as suggested. Accordingly, We edited the text and adhered to the effects of the medicinal herbs on the studied organs (muscles, immune system, and intestine) to achieve the targets of the review context (Table 1 & L 433,434,436, 449-460, L556, L587-592).

We thank you again for your careful review.